# Monocyte Differentiation into Destructive Macrophages on In Vitro Administration of Gingival Crevicular Fluid from Periodontitis Patients

**DOI:** 10.3390/jpm11060555

**Published:** 2021-06-15

**Authors:** Hammam Ibrahim Fageeh, Hytham N. Fageeh, Shankargouda Patil

**Affiliations:** 1Department of Preventive Dental Science, College of Dentistry, Jazan University, Jazan 45412, Saudi Arabia; hafageeh@jazanu.edu.sa (H.I.F.); hfageeh@jazanu.edu.sa (H.N.F.); 2Department of Maxillofacial Surgery and Diagnostic Sciences, Division of Oral Pathology, College of Dentistry, Jazan University, Jazan 45412, Saudi Arabia

**Keywords:** cytokine, gingival crevicular fluid, monocyte, macrophage, periodontitis

## Abstract

Background: Periodontitis is an inflammatory condition of the tooth-supporting structures initiated and perpetuated by pathogenic bacteria present in the dental plaque biofilm. In periodontitis, immune cells infiltrate the periodontium to prevent bacterial insult. Macrophages derived from monocytes play an important role in antigen presentation to lymphocytes. However, they are also implicated in causing periodontal destruction and bystander damage to the host tissues. Objectives: The objective of the present study was to quantify the cytokine profile of gingival crevicular fluid (GCF) samples obtained from patients with periodontitis. The study further aimed to assess if GCF of periodontitis patients could convert CD14+ monocytes into macrophages of destructive phenotype in an in vitro setting. The secondary objectives of the study were to assess if macrophages that resulted from GCF treatment of monocytes could affect the synthetic properties, stemness, expression of extracellular matrix proteins, adhesion molecules expressed by gingival stem cells, gingival mesenchymal stromal cells, and osteoblasts. Methods: GCF, blood, and gingival tissue samples were obtained from periodontitis subjects and healthy individuals based on specific protocols. Cytokine profiles of the GCF samples were analyzed. CD14+ monocytes were isolated from whole blood, cultured, and treated with the GCF of periodontitis patients to observe if they differentiated into macrophages. Further, the macrophages were assessed for a phenotype by surface marker analysis and cytokine assays. These macrophages were co-cultured with gingival stem cells, epithelial, stromal cells, and osteoblasts to assess the effects of the macrophages on the synthetic activity of the cells. Results: The GCF samples of periodontitis patients had significantly higher levels of IFN gamma, M-CSF, and GM-CSF. Administration of the GCF samples to CD14+ monocytes resulted in their conversion to macrophages that tested positive for CD80, CD86, and CD206. These macrophages produced increased levels of IL-1β, TNF-α, and IL-6. Co-culture of the macrophages with gingival stem cells, epithelial cells, and stromal cells resulted in increased cytotoxicity and apoptotic rates to the gingival cells. A reduced expression of markers related to stemness, extracellular matrix, and adhesion namely OCT4, NANOG, KRT5, POSTN, COL3A1, CDH1, and CDH3 were seen. The macrophages profoundly affected the production of mineralized nodules by osteoblasts and significantly reduced the expression of *COL1A1*, *OSX*, and *OCN* genes. Conclusion: In periodontitis patients, blood-derived monocytes transform into macrophages of a destructive phenotype due to the characteristic cytokine environment of their GCF. Further, the macrophages affect the genotype and phenotype of the resident cells of the periodontium, aggravate periodontal destruction, as well as jeopardize periodontal healing and resolution of inflammation.

## 1. Introduction

Periodontitis is an inflammatory condition of microbial etiology that affects the tooth-supporting tissues collectively referred to as the periodontium. This complex multicellular and histologically diverse tissue is composed of the gingiva and the periodontal ligament. The gingiva forms a soft tissue casing. The fibrous periodontal ligament anchors the avascular root cementum to the alveolar bone complex. Poor oral hygiene results in the accumulation of the dental plaque biofilm around the gingival margins. Plaque elicits an inflammatory response in the gingiva that is characterized by an infiltration of this tissue initially by polymorphonuclear leukocytes that predominantly constitute the innate immune response. This is succeeded by the recruitment and movement of monocyte-derived macrophages, T and B lymphocytes as a part of the specific or adaptive immune response [1]. Periodontitis is a progression of inflammation that spreads from the gingival tissues to involve and destroy the underlying periodontal ligament, cementum, and alveolar bone resulting in the formation of periodontal pockets associated with clinically appreciable attachment loss and tooth mobility. This can lead to tooth loss, aesthetic, and functional concerns for the afflicted patient [2]. The host response in periodontitis is also characterized by the excessive production of inflammatory exudate, termed the gingival crevicular fluid (GCF). GCF originates from the vascular network of the gingival corium [3]. GCF samples obtained from periodontitis patients contain excessive amounts of pro-inflammatory cytokines, such as IL-1β [4,5,6] and TNF-α [7,8]. The cytokine-rich GCF release and its accumulation in the periodontium in the state of periodontitis are believed to create a microenvironment that drives the immune and inflammatory cells in the direction of tissue destruction rather than disease resolution.

Among the immune cells that participate in periodontal destruction, the monocytes and macrophages deserve special attention. Monocytes and macrophages are a bridge between innate and adaptive immune responses. They originate from a common progenitor known as monocyte-macrophage DC progenitor (MDP). Monocytes from the blood reach the tissue microenvironment and transform into macrophages [9,10,11]. Macrophages exist in 2 predominant phenotypes namely M1 and M2. M1 subtype is known as the destructive macrophage phenotype. The M2 subtype is known as a reparative macrophage phenotype. The macrophages show positive cell surface expression of CD80, CD86, CD206, and show increased expression of suppressor of cytokine signaling-3 (SOCS3) and inducible nitric oxide synthase (iNOS) based on their phenotype [12,13,14,15]. iNOS, which produces nitric oxide (NO) is a key enzyme for the inflammatory role of macrophages, which predominantly participate in the pathogenesis of inflammatory diseases [12,16]. In periodontitis, an increased proportion of the M1 macrophages have been demonstrated implicating a destructive role played by these cells in periodontal pathogenesis [17]. Increased amounts of monocyte-specific chemokines in the GCF of patients with periodontitis have been recorded in a few studies [8,18].

GCF creates and sustains a cytokine-rich environment in the periodontium that favors a transformation of blood-derived CD14+ monocytes into M1 macrophages. The consequence of increased numbers of M1 macrophages in the periodontium could result in the creation of a long-lasting inflammatory microenvironment. As the macrophages possess chemokine receptors, such as CCR2, CXCR5, CCR5, and CCR1 [19], they are known to migrate along the CCL2 and CCL3 chemokine gradients. They can eventually produce several inflammatory cytokines and chemokines that include CCL2, IL-1β, IL-6, IL-8, and TNF-α, which could cause periodontal tissue destruction [20]. Macrophages have also been hypothesized to interact with the local cellular population in peripheral tissues and modulate the activity of Mesenchymal-origin stromal cells such as fibroblasts and osteoblasts, mesenchymal stem cells (MSCs), and epithelial cells. Macrophages influence stemness, proliferation, apoptosis, synthesis, and degradation of matrix molecules, and expression of cell adhesion molecules.

The present study aimed to quantify the cytokine profile of GCF samples obtained from patients with periodontitis and sought to examine whether GCF of periodontitis patients could convert CD14+ monocytes into macrophages of destructive phenotype in an in vitro setting. The secondary objective of the study was to examine whether macrophages that resulted from GCF treatment of monocytes could affect the synthetic properties, stemness, and expression of extracellular matrix proteins and adhesion molecules expressed by gingival stem cells, gingival mesenchymal stromal cells, and osteoblasts.

## 2. Materials and Methods

Ethical approval: The present study was approved by Scientific Research—College of Dentistry, Jazan University (Reference number: CODJU-2005F). All the participants were informed about the study and formal consent was obtained in the local language.

Inclusion, exclusion criteria, and details of sample collection: GCF samples were collected from 10 patients diagnosed with generalized periodontitis with sites in all 4 quadrants presenting with Stage 3/Stage 4 severity according to the latest guidelines laid down in the 2018 World Workshop on the classification of periodontal diseases [21]. GCF samples, antecubital vein blood samples, and gingival tissue samples were additionally obtained from 10 periodontally and systemically healthy subjects who came for surgical crown lengthening procedure/orthodontic extraction. The exclusion criteria for the study were the presence of systemic diseases, smoking, alcoholism, and substance abuse habit. Pregnant women and lactating mothers were excluded from the study. Age and gender matching were done in the periodontitis and healthy group to eliminate bias in this regard. The workflow of the study has been succinctly presented in the graphical abstract.



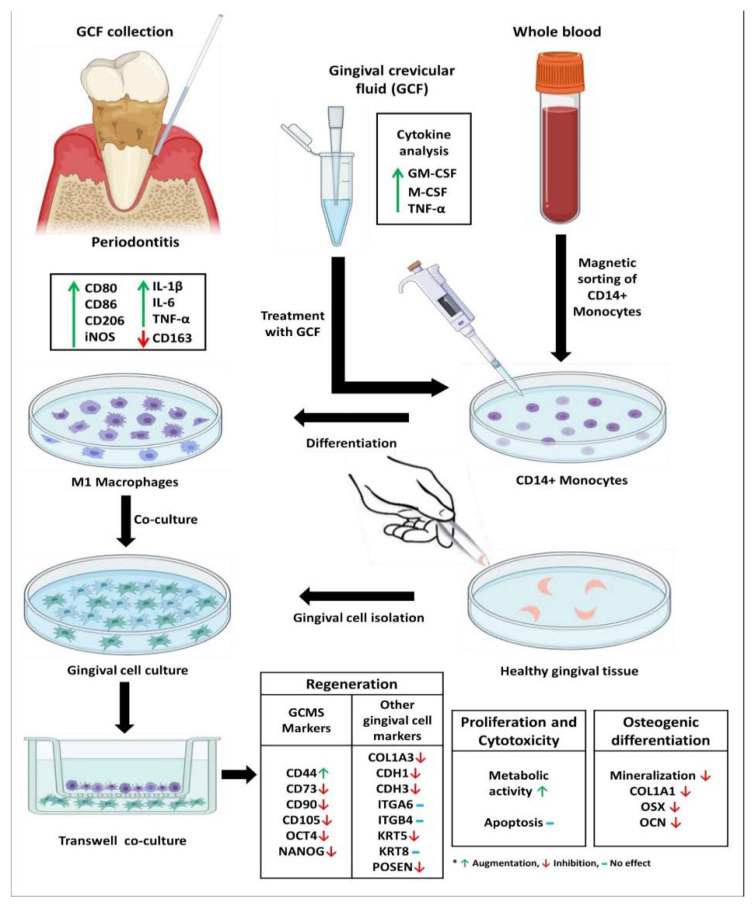



Protocol for GCF, blood collection, and gingival tissue sampling:

GCF sample collection protocol: After a clinical examination, the patients were seated upright in the dental chair with good illumination. GCF was sampled from the sites with Stage 3 or Stage 4 periodontitis in each quadrant of the patient’s mouth using microcapillary pipettes under a standard protocol [22]. Isolation before sample collection was done using cotton rolls and high-volume suction to prevent saliva contamination. GCF samples contaminated with blood were discarded and not used for further analysis. The GCF collected from the 4 sites in each patient was pooled together and centrifuged at 300 xg and stored at −80 °C until further analysis was done. The same sampling protocol was followed for the systemically and periodontally healthy subjects to obtain pooled GCF samples from 4 periodontally healthy sites. The GCF samples obtained from the control subjects were used for cytokine analysis to compare the levels of cytokines in the GCF samples of periodontitis patients. One portion of the sampled GCF from each periodontitis patient was used to analyze the cytokine profile, while the other portion was used for in vitro CD14+ treatment.

Blood sample collection: A 20 mL blood sample was collected from the antecubital vein using a sterile aseptic protocol. The blood sample was immediately transported to the molecular biology facility for isolation of CD14+ monocytes and culture of the cells. The GCF sample collected from a periodontitis patient was used to stimulate the CD14+ monocytes obtained and cultured from the systemically and periodontally healthy subjects.

Gingival tissue sample collection: Gingival tissue samples were obtained from the healthy subjects as previously mentioned by gingivectomy under local anesthesia using a Bard parker 15 size blade mounted on a handle using a sterile and aseptic protocol. The tissues collected were macerated, homogenized, and subjected to the explant culture method to isolate and culture gingival mesenchymal stem cells, and stromal cells.

Cytometric bead array for the estimation of cytokine levels in the GCF samples: a Cytometric bead array was performed to assess the cytokine levels in the GCF. LEGENDplex™ Human Essential Immune Response Panel was utilized for cytokine detection of selected cytokines related to monocytes and macrophages namely IFN-γ, GM-CSF, and M-CSF. The experimental protocol was implemented as per the manufacturer’s instructions. 25 μL of the GCF samples were incubated for 2 h with microbeads. Following incubation, detection antibodies were added and incubated for thirty minutes. After incubation, the samples were washed with a wash buffer. The samples were centrifuged at 2000 RPM for 5 min. Following the supernatant elimination, the pellet was resuspended in a 200 μL sheath fluid. A flow cytometer was used to acquire the samples (Attune NxT, Thermo Fisher Science, Waltham, MA, USA). Analysis of the obtained data was done using LEGENDplex™ Data Analysis Software (BioLegend, San Diego, CA, USA) to quantify cytokine levels using known standards.

Isolation of peripheral blood monocytes, and sorting of CD14+ monocytes by magnetic-activated cell sorting and flow cytometric analysis of CD14+ cells: The mononuclear cells of the peripheral blood (PBMCs) were separated from blood by the centrifugation method of density gradient by utilizing Histopaque-1077 (Sigma, St. Louis, MO, USA). The isolated PBMCs were washed three times with phosphate-buffered saline. Magnetic sorting of human peripheral blood-derived CD14+ monocytes with CD14 (anti-human) MicroBeads (Miltenyi Biotec, Bergisch Gladbach, Germany), was done by adhering to the manufacturer’s standard operating procedures. The sorted CD14+ cells were labeled with Anti-CD14-APC antibody (Miltenyi Biotec, Bergisch Gladbach, Germany). The cells were checked for purity by performing flow cytometric analysis before seeding the cells with complete media (DMEM + 10% FBS) for further experimentation.

Cell culture and treatment with GCF: CD14+ monocytes were plated in 12-well cell culture plates at a seeding density of 5 × 10^5^ cells per well and incubated with a complete medium (DMEM + 10% FBS). Each well was treated with 2 µL GCF/mL. After 48 h of incubation, the media along with the floating cells was discarded from each well and topped up with a fresh complete medium. The treatment was repeated at each media change for 7 days. The untreated CD14+ monocytes were regarded as the control.

Analysis of macrophage cell surface marker expression of CD14, CD80, CD86, CD163, and CD206 by flow cytometry: The differentiated cells were removed from the culture plates of GCF treated CD14+ monocytes and incubated with macrophage markers CD14, CD80, CD86, CD163, and CD206 antibodies (Miltenyi Biotec, Bergisch Gladbach, Germany) respectively. After the 30-min room temperature incubation, the antibody-stained cells were analyzed on a flow cytometer (Attune NxT, Thermo Fisher Scientific, Waltham, MA, USA). At least 10,000 cell events were gated for an individual sample. The percentage of positive cells was calculated for comparative analysis in each group.

Real-time quantitative polymerase chain reaction (RT-qPCR) for quantitative analysis of *iNOS* gene expression in the CD14+ monocytes and macrophages: Total RNA was isolated from the pelleted down cells from the CD14+ untreated monocytes and the differentiated macrophages using the GeneJET RNA Purification Kit (Thermo Scientific, Vilnius, Lithuania), following the manufacturer’s instructions. The quality analysis and concentration of RNA were done on a NanoDrop spectrometer. Synthesis of cDNA was performed with the cDNA Reverse Transcription Kit (High Capacity, Applied Biosystems, Carlsbad, CA, USA) and diluted for RT-qPCR as recommended by the manufacturer. The amplification of cDNA was carried out with SYBR Green Master Mix (Applied Biosystems, Austin, TX, USA) for genes of interest. Appropriate primers of the *iNOS* gene were used (IDT, Coralville, IA, USA). GAPDH served as the housekeeping gene. The cycle threshold (CT) values for each gene were corrected by using the mean CT value. mRNA levels were calculated by the ΔΔCt method and were quantified by using the 2^−ΔΔCt^ method, normalized to the average CT for the GAPDH gene expression levels. The primer sequences used are shown in Table 1.

ELISA for analysis of cytokine levels of TNF-α, IL-6, and IL-1β in the conditioned media of the CD14+ monocytes and macrophages: The analysis of TNF-α, IL-6, and IL-1β at the protein level was carried out by using KRIBIOLISA human ELISA kits (Krishgen Biosystems, Los Angeles, CA, USA) in the GCF untreated CD14+ monocytes and differentiated macrophage cultures. The conditioned media from CD14+ monocytes and macrophages were diluted 10 times, and the protocol was performed conferring to the experimental instructions provided with the kit. A spectrophotometer was used to record the absorbance at 450 nm.

Enzymatic digestion/explant culture of gingival tissue for cell isolation and cell culture: The gingival tissues obtained from systemically and periodontally healthy subjects were rinsed with an antibiotic antimycotic containing sterile phosphate-buffered saline (PBS) (Sigma, St. Louis, MO, USA). A sterile blade was used to mince the tissues further. Following this, they were exposed directly to enzymatic digestion using Dispase II (Roche Diagnostics GmbH, Mannheim, Germany) and Collagenase I (MP Biomedicals LLC, Santa Ana, CA, USA) solutions. In the enzyme mixture, the minced tissue was incubated for an hour at 37 °C. The digested tissue was passed through a sterile cell strainer of 70 μm in pore size (Corning, NY, USA) after countering the enzyme action with fetal bovine serum (FBS). The flowthrough was then centrifuged for 5 min at 2000 RPM. Finally, the pellets were resuspended in a complete cell culture medium (DMEM + 10% FBS) (Invitrogen, Carlsbad, CA, USA) and incubated at 37 °C under a 5% CO_2_ atmosphere. The explant culture method was used to isolate and culture gingival stem cells (GMSCs). The cells that grew out of this experiment represented a heterogeneous population of gingival epithelial cells, fibroblasts, and gingival stem cells. A portion of these cells was left untreated, while the other portion was co-cultured with differentiated macrophages according to a standard protocol

Surface marker analysis of gingival cells in control versus co-culture group by flow cytometry: The untreated gingival cells and the gingival cells co-cultured with macrophages were separated from the surface of the culture flasks and resuspended in PBS and divided into different tubes. Each tube was incubated with CD44-PE, CD73-PE, CD90-FITC, and CD105-APC antibodies for 30 min at 4 °C (Miltenyi Biotec, Bergisch Gladbach, Germany). The cells were washed with PBS following incubation and pelleted down. The labeled cells were resuspended in PBS and analyzed on a flow cytometer (Attune NxT, Thermo Fisher Scientific, Waltham, MA, USA). At least 10,000 events per sample were acquired. In comparison with the isotypic controls, the degree of positive staining was measured as a percentage.

MTT assay to assess cell viability (3-(4,5-dimethylthiazol-2-yl)-2,5-diphenyltetrazolium bromide) in untreated gingival cells versus gingival cells co-cultured with macrophages: The viability of cells was assessed by performing an MTT assay. The cells were seeded into 96-well plates at a cell density of 5 × 10^4^ cells per well. Growth media was used for the incubation of the cells for 24 h. After co-culturing them with macrophages in a transwell co-culture system for 24 h, 0.5 mg/mL MTT solution was introduced to each well. These plates were incubated for 3 h. Finally, in each well, the medium was replaced with 100 µL of dimethyl sulfoxide (DMSO). The absorbance was read at 570 nm on a spectrophotometer (Multiskan Spectrum, Thermo Scientific, San Jose, CA, USA).

Apoptosis analysis using Annexin-PI staining by flow cytometry in untreated gingival cells versus gingival cells co-cultured with macrophages: Apoptosis detection assay was performed using Annexin-PI staining. The cells from each group were harvested after 24 h of incubation in the wells of a 12-well plate seeded at the density of 5 × 10^4^ cells per well and stained with Annexin-PI-FITC reagent. After incubation for 30 min, the cells were directly acquired with at least 10,000 events per sample on a flow cytometer (Attune NxT, Thermo Fisher Scientific) and the percentage of apoptotic cells was calculated by using Attune NxT Software analysis.

Real-time quantitative polymerase chain reaction (RT-qPCR) for quantitative analysis of gene expression in untreated gingival cells versus gingival cells co-cultured with macrophages: Total RNA was isolated from the pelleted down cells in the untreated group versus the gingival cells co-cultured with macrophages using the GeneJET RNA Purification Kit (Thermo Scientific, Lithuania), by following the instruction of the manufacturer’s protocol. The quality analysis and concentration of RNA were done on a NanoDrop spectrometer. Synthesis of cDNA was performed with the cDNA Reverse Transcription Kit (High Capacity, Applied Biosystems, Carlsbad, CA, USA) and diluted for RT-qPCR as suggested in the manufacturer’s instructions. The amplification of cDNA was carried out with SYBR Green Master Mix (Applied Biosystems, Austin, TX, USA) for genes of interest. Appropriate primers of the *OCT4*, *NANOG*, *KRT5*, *KRT8*, *POSTN*, *COL1A3*, *CDH1*, *CDH3*, *ITGA6*, and *ITGB4* genes were used (IDT, Coralville, IA, USA). GAPDH served as a housekeeping gene. The cycle threshold (CT) values for each gene were corrected using the mean CT value. mRNA levels were calculated by the ΔΔCt method and were quantified using the 2^−ΔΔCt^ method, normalized to the average CT for the *GAPDH* gene expression levels. The primer sequences used are listed in Table 1.

Differentiation of gingival stem cells into osteoblasts and co-culture experiments: Osteogenic differentiation was induced in GMSCs by StemMACS™ OsteoDiff Media, human (Miltenyi Biotec, Bergisch Gladbach, Germany). On day 14 of induction, the differentiated osteoblasts were co-cultured with M1 macrophages for 24 h and subjected to functional staining with alizarin red S, quantification of mineralization. The differentiated cells were fixed with 4% paraformaldehyde, and 2% alizarin red S (pH 4.1–4.3) staining was performed for 20 min. The quantification of alizarin red S-stained osteoblasts was done by dissolving stained cells in 4% NaOH, and the dissolved stain was read spectrophotometrically at 450 nm. Gene expression analysis, similar to the above-mentioned protocol, was done to quantify the *COL1A1*, *OSX*, and *OCN* genes. The primer sequences used are listed in Table 1. Osteoblasts not subjected to the co-culture were regarded as controls.

Statistical analysis: All the experimental results were shown as the mean ± standard deviation (SD) of the values from the three independent experimental values as the experiments were conducted in triplicates. The data were assessed by using unpaired Students *t*-test (two-tailed) on GraphPad Prism software. A *p*-value < 0.05 was measured as significant and *p*-value < 0.01 was measured as highly significant (* *p* < 0.05 and ** *p* < 0.01), while *p* > 0.05 was regarded as non-significant.

## 3. Results

The results of the study are depicted below under the relevant subheadings.

GCF analysis of periodontitis patients versus healthy subjects: The GCF samples obtained from the periodontitis patients had significantly higher levels of IFN-γ, GM-CSF, and M-CSF compared to the periodontally and systemically healthy subjects. The graphical data are presented in Figure 1A–C.

Isolation of CD14+ monocytes from peripheral blood of healthy subjects and macrophage differentiation induction by GCF treatment: CD14+ positive monocytes were isolated from the peripheral blood monocytes. It was found that there was a statistically significant upregulation of CD80, CD86, and CD206 and significant downregulation of CD163 in the isolated macrophages following GCF treatment when compared to the untreated CD14+ monocytes. There was no significant difference in CD14 expression between macrophages following GCF treatment when compared to the untreated CD14+ monocytes. Figure 2 depicts the flow cytometry histograms, while Figure 3A–E present the graphical data.

Gene and protein analysis of untreated CD14+ monocytes and isolated macrophages following GCF treatment: A statistically significant elevation in the *iNOS* gene expression in isolated macrophages following GCF treatment compared to untreated CD14+ monocytes (data in Figure 4A) was found. Protein analysis of the conditioned media revealed significantly higher levels of TNF-α, IL-6, and IL-1β in isolated macrophages following GCF treatment compared to untreated CD14+ monocytes (data presented in Figure 4B–D).

Isolation of stem cells and heterogeneous mesenchymal cell populations from the gingiva, and subsequent co-culture experiments with GCF treatment-induced macrophages to assess cell viability, apoptosis, and cell surface mesenchymal stem cell markers: The gingival tissue explant culture technique yielded a heterogeneous population of gingival mesenchymal stem cells and other stromal cells. These mixed populations of cells were co-cultured with the macrophages that were induced by GCF treatment of CD14+ monocytes. The MTT assay revealed a significant increase in metabolic activity in terms of cytotoxicity in co-cultured cells versus control gingival cells (data presented in Figure 5A).

The data was corroborated with the apoptosis experiments that revealed an increase in apoptosis in co-cultured cells versus control gingival cells, although not statistically significant (*p* > 0.05) (data in Figure 5B). Co-culture with macrophages significantly reduced the surface expression of CD73, CD90, and CD105, which are markers of mesenchymal stem cells, and significantly increased the expression of CD44 in terms of median fluorescence intensity. The numerical data are presented in Figure 6A–H.

Gene expression of potential stemness markers, cytoskeletal proteins, epithelial extracellular matrix proteins, and cell adhesion molecule in untreated gingival cells versus gingival cells co-cultured with macrophages: Gene expression analysis was done on a wide range of genes, including stemness (OCT4 and NANOG), keratinocyte cytoskeletal proteins (KRT5 and KRT8), epithelial extracellular matrix proteins (COL3A1 and POSTN), epithelial cell-cell adhesion molecules (CDH1 and CDH3), and cell-surface adhesion molecules (ITGA6 and ITGB4) in untreated gingival cells versus gingival cells co-cultured with macrophages. Results revealed a significant downregulation of OCT4 and NANOG. *KRT5*, *POSTN*, *COL3A1*, *CDH1*, *CDH3* genes in the co-culture versus control cells. There was no significant difference observed between the 2 groups concerning *KRT8*, *ITGA6*, and *ITGB4* gene expression. The relevant statistical data are presented in Figure 7A–J.

Differentiation of gingival stem cells into osteoblasts and co-culture experiments to detect mineralization and bone metabolism-related genes: The gingival tissue explant culture technique yielded a heterogeneous population of gingival mesenchymal stem cells and other stromal cells. These cells were differentiated into osteoblasts by well-defined protocols. The osteoblasts were co-cultured with macrophages and subjected to assessment of mineralized nodules, gene profiling, and quantification of proteins related to bone metabolism. Osteoblasts not subjected to co-culture were regarded as control. Results revealed a significant reduction in mineralized nodules and expression of the *COL1A1*, *OSX*, and *OCN* genes in the co-culture group compared to the control group. The graphical data are presented in Figure 8A–F.

## 4. Discussion

The present study was conducted to elucidate the role played by macrophages in the pathobiology of periodontitis. The initial analysis of the GCF samples revealed an increase in the levels of IFN-γ, M-CSF, and GM-CSF in periodontitis patients as compared to the healthy subjects. These results are concurrent with previous studies that have demonstrated a similar trend [23,24,25]. The GCF samples obtained from periodontitis patients were used to treat CD14+ monocytes obtained from peripheral blood of healthy donors. CD14+ monocytes were found to differentiate into macrophages, which significantly overexpressed the surface markers CD80, CD86, and CD206. The macrophages under-expressed CD163 compared to the untreated CD14+ monocytes. These findings imply the monocytes differentiated into a mixed population of macrophages of both the M1 and M2 phenotypes. Previous studies have documented that CD80 and CD86 are costimulatory receptors predominantly expressed on macrophages that trigger intensive inflammation, and cause tissue destruction [26]. CD206, referred to as mannose receptor-1, is a marker of macrophages contributing to repair [27]. A clear finding in the present study was the downregulation of CD163 in the macrophages. This finding is significant in the current context as CD163 is a marker of the anti-inflammatory macrophage [28]. A low expression of this marker indicates that GCF treatment of monocytes differentiates them predominantly into classical M1 macrophages with a destructive phenotype. To reconfirm this finding at a genetic and protein level, we assessed the expression of the *iNOS* gene, the levels of TNF-α, IL-6, and IL-1β protein by quantitative PCR and ELISA technique in the macrophages and untreated CD14+ monocytes. A significant increase in *iNOS* gene expression in the macrophages was observed compared to the untreated monocytes. iNOS is an enzyme involved in the synthesis of nitric oxide, which is an important reactive nitrogen species. Previous studies have demonstrated that nitric oxide levels are increased in periodontitis sites with active disease [29,30,31]. Our findings confirm that the phenomenon responsible for the increased nitric oxide levels could be the M1 macrophage-mediated iNOS production. The macrophage conditioned media had significantly higher levels of TNF-α, IL-1β, and IL-6 compared to untreated monocytes. These findings confirm that the macrophages resulting from GCF treatment of monocytes have the genotype and phenotype characteristics of the classical M1 macrophages.

The present study also aimed to assess the paracrine effects of the macrophages on gingival stromal cells, stem cells, and osteoblasts in a co-culture model. Gingival tissues were collected from healthy donors, and the explant culture method was used to culture a mixed population of gingival mesenchymal stem cells, stromal cells containing a loose population of epithelial cells, and fibroblasts. The presence of gingival mesenchymal stem cells in this heterogeneous cell mixture was confirmed by fluorescent-activated cell sorting to detect the mesenchymal stem cell markers. The cultures significantly expressed CD73, CD90, and CD105, which are mesenchymal stem cell markers [32]. There was a low but detectable expression of CD44, an important activator of leukocytes [33]. These findings were reversed when the gingival cells were co-cultured with the macrophages. A significant lowering in CD73, CD90, and CD105 expression was observed along with upregulated CD44. This could be a result of cytotoxicity of the macrophage products on the stem cells, consequently causing apoptosis of the stem cells, and cell death as confirmed by MTT cytotoxicity assay, and apoptosis assay with annexin 1.

To further assess the effects of the macrophage co-culture on various markers in the heterogeneous gingival cell population, a quantitative PCR analysis was performed. Analysis revealed a significant downregulation of markers of stemness, cell adhesion, and extracellular matrix, namely OCT4 and NANOG, KRT5, POSTN, COL3A1, CDH1, and CDH3 in the cells following co-culture compared to the control cells. Macrophages have a profound paracrine effect on the epithelial and connective tissue compartments of the gingiva and could affect the healing mechanism as these markers reflect the healthy homeostatic state of the periodontium.

Finally, an assessment of the effect of macrophages on the synthetic processes of osteoblasts was carried out. The gingival stem cells that resulted from the explant technique were differentiated into osteoblasts. The cells were co-cultured with macrophages and subjected to alizarin red staining, and PCR analysis to quantify the *COL1A1, OSX*, and *OCN* genes. The results revealed that in the co-culture group, there was a significant reduction in the expression of the above genes coupled with a reduction in the number of mineralized nodules when compared to osteoblasts that were not subjected to co-culture. The findings of macrophage osteoblast interaction in the present study hint that the macrophages are polarized towards the M1 phenotype. A similar situation is encountered in the pathogenesis of osteoarthritis, in which macrophage polarization causes a predominance of M1 macrophages. This results in the release of pro-inflammatory cytokines into the synovial microenvironment, causing osteoblast dysfunction, and inducing osteophyte formation, joint swelling, inflammation, and pain [34].

The results indicate that macrophages are the predominant immune cells that regulate the pathogenesis of periodontitis. A sustained release of inflammatory cytokines and mediators by the macrophages perpetuates a vicious cycle of periodontal destruction. Macrophages appear to affect the synthetic properties of epithelial cells, stromal cells, mesenchymal stem cells, and osteoblasts. This could explain the reduced healing of periodontal lesions.

## 5. Conclusions

Our findings elucidate the role of macrophages in the pathogenesis of periodontitis. Monocytes transform into macrophages of a destructive phenotype due to the characteristic cytokine environment of their GCF. Data from the present study can be utilized to develop host modulatory therapeutic strategies in periodontitis management based on targeting vital immune cells such as the macrophage. Using these data, innovative treatment strategies, such as the use of sub-antimicrobial dose doxycycline [35] and resolvins [36], can be refined and implemented for periodontal disease management.

## Figures and Tables

**Figure 1 jpm-11-00555-f001:**
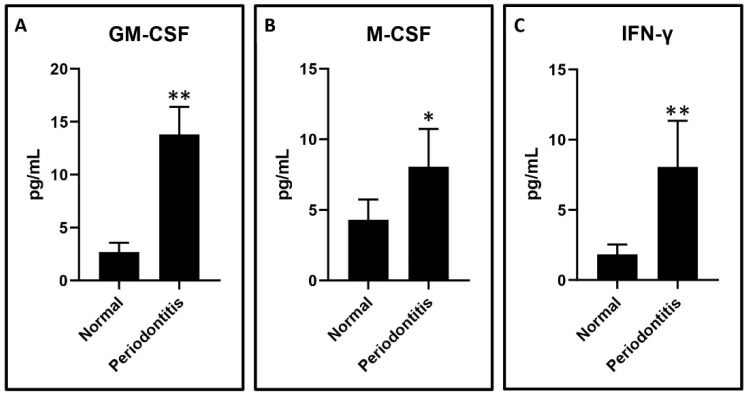
GCF analysis of periodontitis patients versus healthy subjects. (**A**)—GM-CSF (** *p* < 0.0001); (**B**)—M-CSF (* *p* = 0.0238); (**C**)—IFN-γ (** *p* < 0.0001).

**Figure 2 jpm-11-00555-f002:**
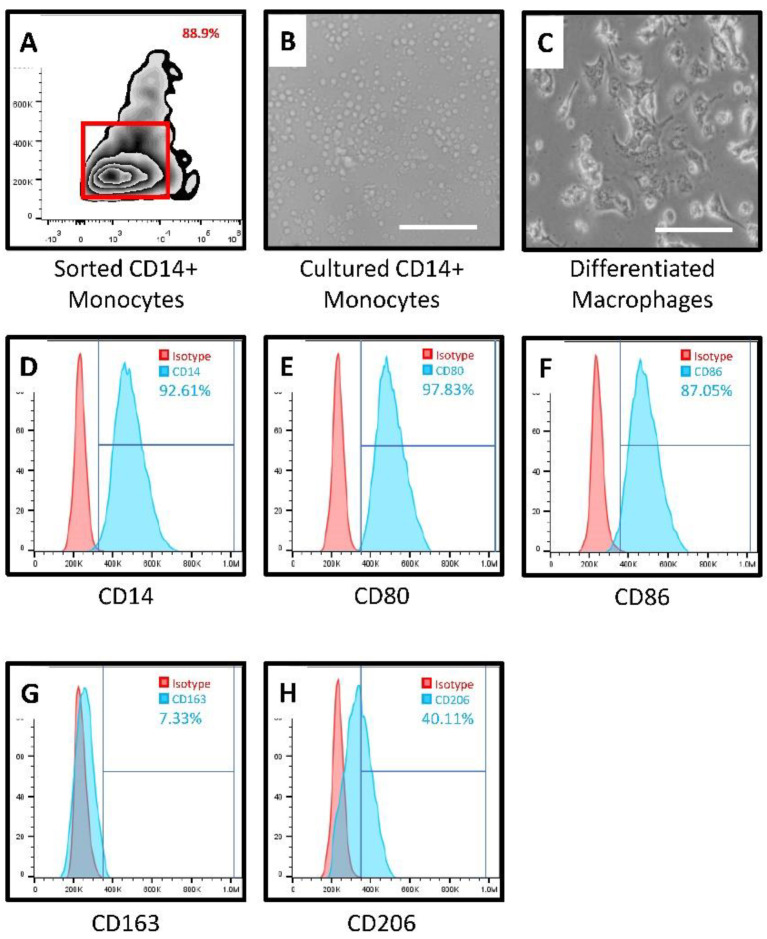
(**A**)—CD14+ monocyte sorting; (**B**)—Cultured CD14+ monocytes; (**C**)—GCF treated CD14+ monocytes differentiating into macrophages; Cell surface analysis of the differentiated macrophages with (**D**)—CD14, (**E**)—CD80, (**F**)—CD86, (**G**)—CD163, (**H**)—CD206 respectively.

**Figure 3 jpm-11-00555-f003:**
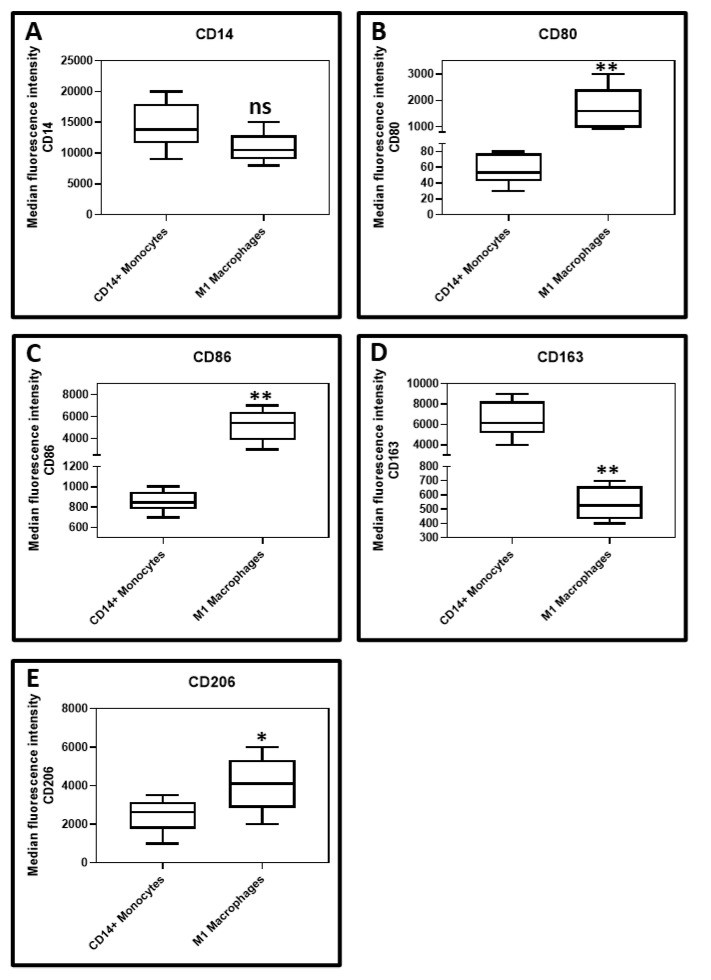
Comparative cell surface analysis between GCF differentiated macrophages and untreated CD14+ monocytes for (**A**)—CD14 (*p* = 0.0551), (**B**)—CD80 (** *p* < 0.0001), (**C**)—CD86 (** *p* < 0.0001), (**D**)—CD163 (** *p* < 0.0001), (**E**)—CD206 (* *p* = 0.0455) respectively.

**Figure 4 jpm-11-00555-f004:**
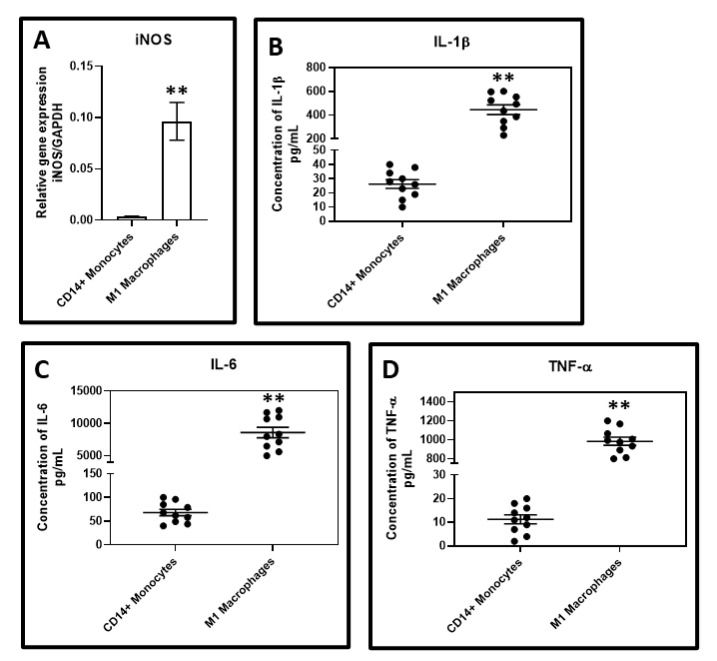
(**A**)—Comparative gene expression analysis between GCF differentiated macrophages and untreated CD14+ monocytes for *iNOS* gene expression (** *p* < 0.0001). Comparative secretome analysis between GCF differentiated macrophages and untreated CD14 monocytes for (**B**)—IL-1β (** *p* < 0.0001), (**C**)—IL-6 (** *p* < 0.0001), (**D**)—TNF-α (** *p* < 0.0001) respectively.

**Figure 5 jpm-11-00555-f005:**
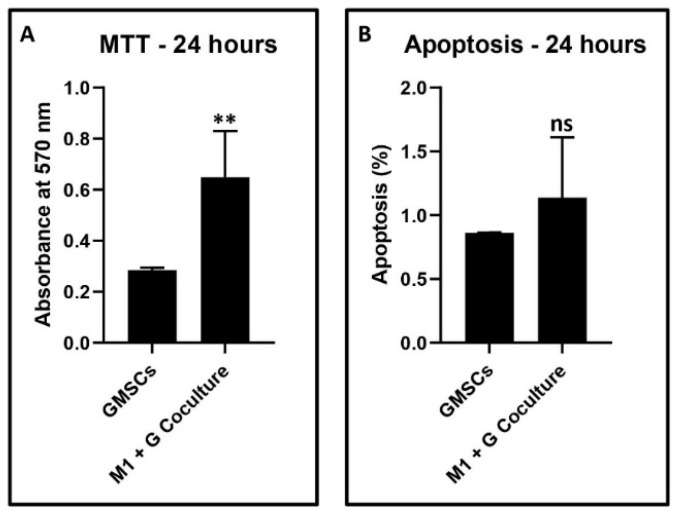
Effect of the GCF differentiated macrophages on gingival cells assessed by (**A**)—MTT assay (** *p* < 0.0001) and (**B**)—Annexin V assay (*p* = 0.0819). ns not significant.

**Figure 6 jpm-11-00555-f006:**
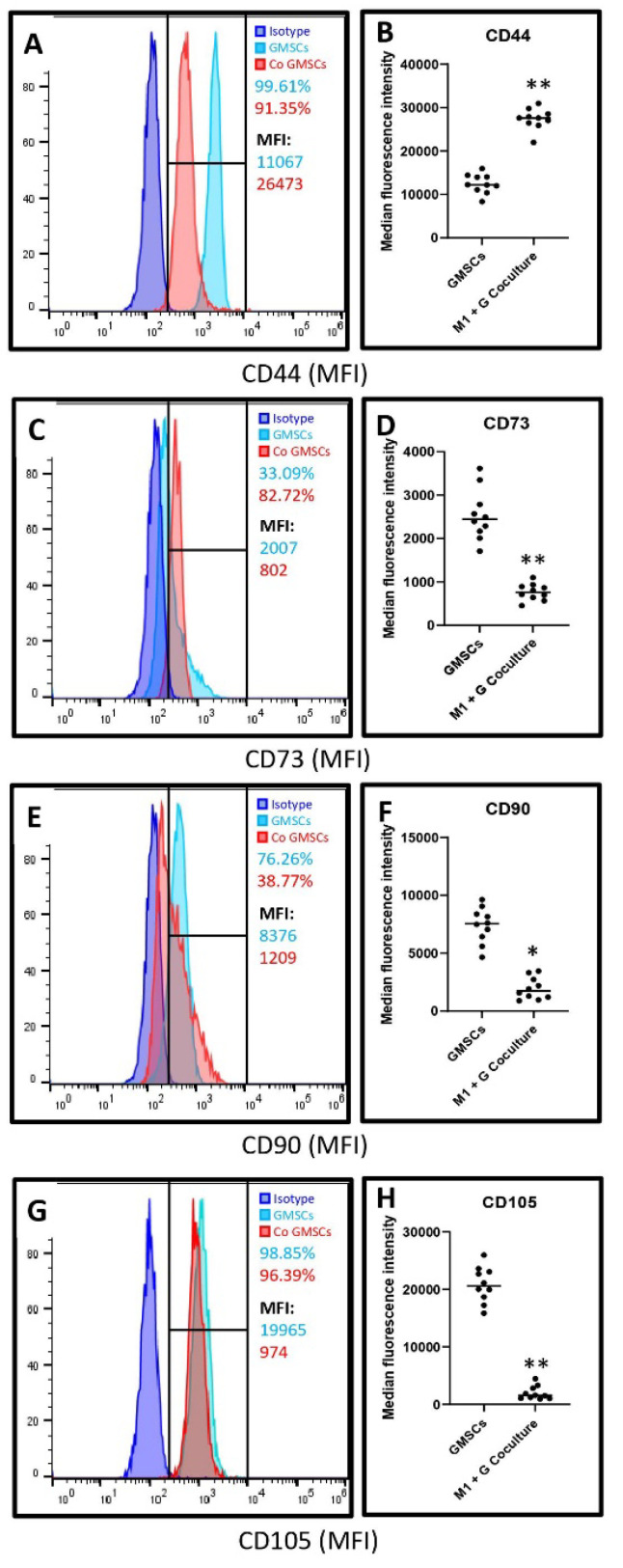
Histograms and comparative analysis depicting (**A**,**B**)—CD44 (** *p* < 0.0001), (**C**,**D**)—CD73 (** *p* < 0.0001), (**E**,**F**)—CD90 (* *p* = 0.0456), and (**G**,**H**)—CD105 (** *p* < 0.0001) staining intensity (median fluorescence intensity) in gingival stem cells and heterogeneous cell populations versus the gingival stem cells and heterogeneous cells co-cultured with macrophages.

**Figure 7 jpm-11-00555-f007:**
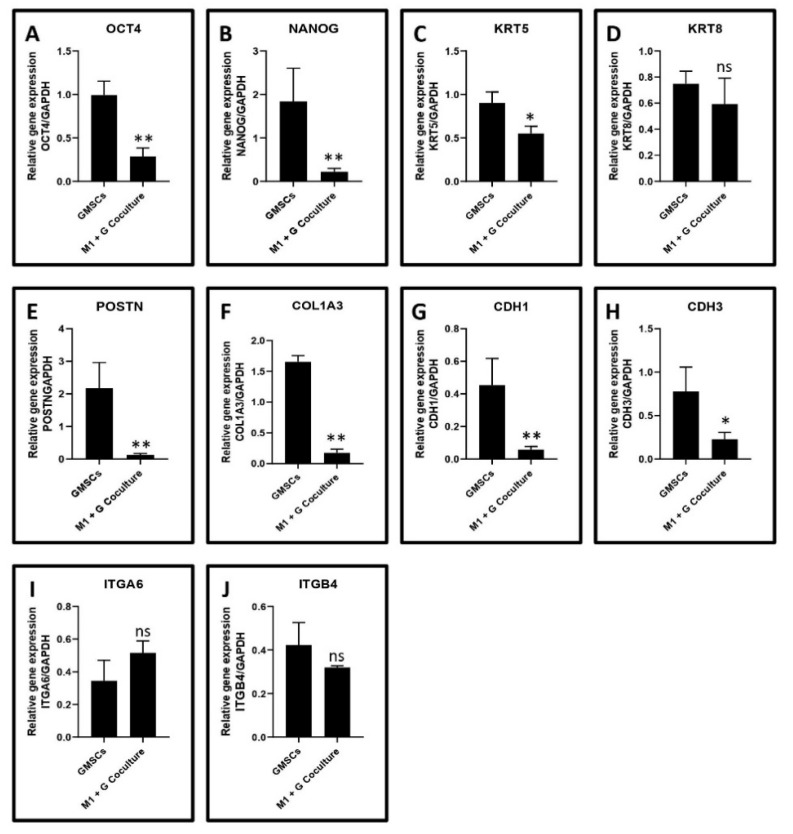
Bar diagrams depicting (**A**)—OCT4 (** *p* < 0.0001), (**B**)—NANOG (** *p* < 0.0001), (**C**)—KRT5 (* *p* < 0.0001), (**D**)—KRT8 (*p* = 0.1599), (**E**)—POSTN (** *p* < 0.0001), (**F**)—COL1A3 (** *p* < 0.0001), (**G**)—CDH1 (** *p* < 0.0001), (**H**)—CDH3 (* *p* = 0.0150), (**I**)—ITGA6 (*p* = 0.0517), (**J**)—ITGB4 (*p* = 0.0995) gene expression in gingival stem cells and heterogeneous cell populations versus the gingival stem cells and heterogeneous cells co-cultured with macrophages. ns not significant.

**Figure 8 jpm-11-00555-f008:**
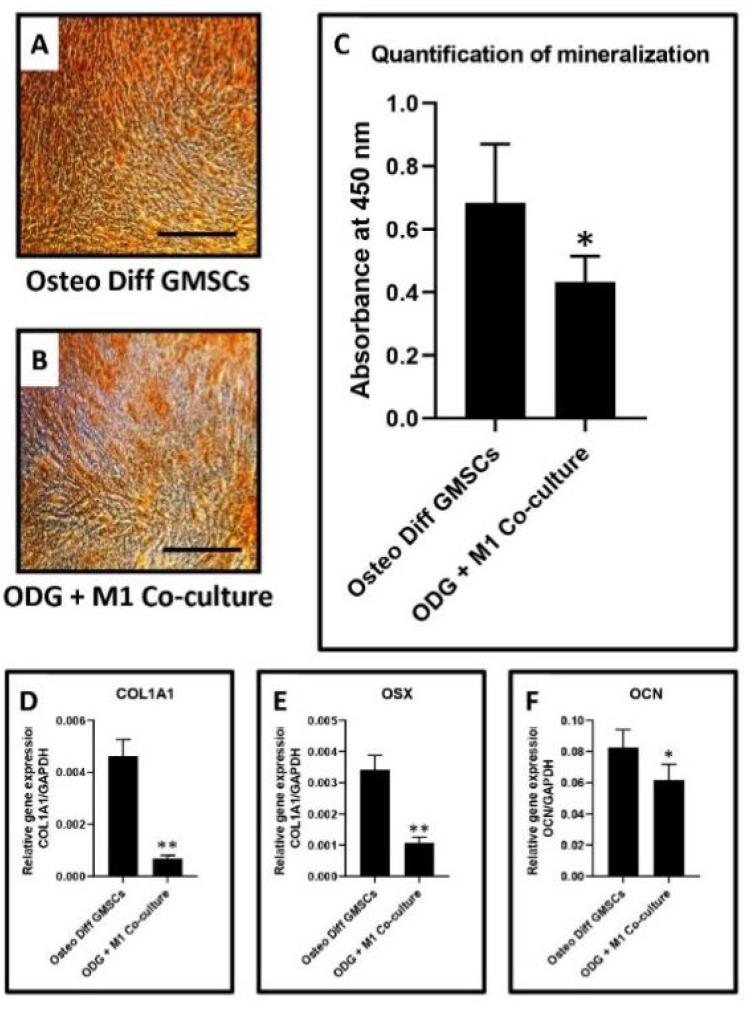
(**A**,**B**)—Alizarin red S staining of osteoblasts derived from gingival stem cells versus osteoblasts derived from gingival stem cells co-cultured with macrophages. (**C**)—Bar diagram depicting mineralization (* *p* = 0.0246) and expression of the (**D**)—COL1A1 (** *p* < 0.0001), (**E**)—OSX (** *p* < 0.0001), (**F**)—OCN (* *p* = 0.0151) genes in osteoblasts derived from gingival stem cells versus osteoblasts derived from gingival stem cells co-cultured with macrophages.

**Table 1 jpm-11-00555-t001:** List of primers.

Gene	Forward Primer	Reverse Primer
*iNOS*	5′-GCT CTA CAC CTC CAA TGT GAC C-3′	5′-CTG CCG AGA TTT GAG CCT CAT G-3′
*OCT4*	5′-CCT GAA GCA GAA GAG GAT CAC C-3′	5′-AAA GCG GCA GAT GGT CGT TTG G-3′
*NANOG*	5′-CTC CAA CAT CCT GAA CCT CAG C-3′	5′-CGT CAC ACC ATT GCT ATT CTT CG-3′
*KRT5*	5′-GCT GCC TAC ATG AAC AAG GTG G-3′	5′-ATG GAG AGG ACC ACT GAG GTG T-3′
*KRT8*	5′-ACA AGG TAG AGC TGG AGT CTC G-3′	5′-AGC ACC ACA GAT GTG TCC GAG A-3′
*POSTN*	5′-CAG CAA ACC ACC TTC ACG GAT C-3′	5′-TTA AGG AGG CGC TGA ACC ATG C-3′
*COL3A1*	5′-TGG TCT GCA AGG AAT GCC TGG A-3′	5′-TCT TTC CCT GGG ACA CCA TCA G-3′
*CDH1*	5′-GCC TCC TGA AAA GAG AGT GGA AG-3′	5′-TGG CAG TGT CTC TCC AAA TCC G-3′
*CDH3*	5′-CAG GTG CTG AAC ATC ACG GAC A-3′	5′-CTT CAG GGA CAA GAC CAC TGT G-3′
*ITGA6*	5′-CGA AAC CAA GGT TCT GAG CCC A-3′	5′-CTT GGA TCT CCA CTG AGG CAG T-3′
*ITGB4*	5′-AGG ATG ACG ACG AGA AGC AGC T-3′	5′-ACC GAG AAC TCA GGC TGC TCA A-3′
*COL1A1*	5′-GAT TCC CTG GAC CTA AAG GTG C-3′	5′-AGC CTC TCC ATC TTT GCC AGC A-3′
*OCN*	5′-GGC GCT ACC TGT ATC AAT GG-3′	5′-TCA GCC AAC TCG TCA CAG TC-3′
*OSX*	5′-TGC TTG AGG AGG AAG TTC AC-3′	5′-AGG TCA CTG CCC ACA GAG TA-3′
*GAPDH*	5′-GTC TCC TCT GAC TTC AAC AGC G-3′	5′-ACC ACC CTG TTG CTG TAG CCA A-3′

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
