# Peer review of "Monocyte Differentiation into Destructive Macrophages on In Vitro Administration of Gingival Crevicular Fluid from Periodontitis Patients"

_jpm, 2021, doi:10.3390/jpm11060555_

Round 1

Reviewer 1 Report

This manuscript is well investigated and the contents are correct and most updated. The manuscript clearly explains the role of macrophages in the pathogenesis of periodontitis and describes how macrophages of destructive phenotype derived from monocytes due to surrounding cytokine environment of Gingival Crevicular Fluid (GCF) contributes to the inflammatory milieu in periodontitis. The strength of this study is that it describes the various aspects of the inflammatory environment in the periodontitis condition. First, it looked at the cytokine profiling of GCF samples and demonstrated an increased levels of inflammatory cytokines. Then, an in-vitro analysis of CD +14 monocytes treated with GCF samples from periodontitis patients was performed, which revealed that the GCF from periodontitis patients could convert CD 14 + monocytes into destructive phenotype macrophages. Another strength of this study was that it assessed the impact of macrophages of destructive phenotype on the genotype and phenotype of the resident cells of the periodontium and how it aggravates periodontal destruction and impacts periodontal healing and resolution of inflammation. The limitation of this manuscript is that it lacks lots of references throughout the manuscript and also there are some minor English language correction are needed.

Author Response

Reviewer 1:

  1. This manuscript is well investigated and the contents are correct and most updated. The manuscript clearly explains the role of macrophages in the pathogenesis of periodontitis and describes how macrophages of destructive phenotype derived from monocytes due to surrounding cytokine environment of Gingival Crevicular Fluid (GCF) contributes to the inflammatory milieu in periodontitis. The strength of this study is that it describes the various aspects of the inflammatory environment in the periodontitis condition. First, it looked at the cytokine profiling of GCF samples and demonstrated an increased levels of inflammatory cytokines. Then, an in-vitro analysis of CD +14 monocytes treated with GCF samples from periodontitis patients was performed, which revealed that the GCF from periodontitis patients could convert CD 14 + monocytes into destructive phenotype macrophages. Another strength of this study was that it assessed the impact of macrophages of destructive phenotype on the genotype and phenotype of the resident cells of the periodontium and how it aggravates periodontal destruction and impacts periodontal healing and resolution of inflammation. The limitation of this manuscript is that it lacks lots of references throughout the manuscript and also there are some minor English language correction are needed.

Response: respected reviewer, we have updated references in the revised manuscript and also done the appropriate language editing wherever nessecary.

Reviewer 2 Report

General comment:

From the title, you expect a lot from this work, but honestly, I found a lot of confusion! It is an interesting subject to treat and the authors need to correct the manuscript as it should, then to be considered for publication. Please find the critical comments as follows:

Critical comments:

Abstract
Lines 16-22: the objectives are not described properly. It is necessary a new reformulation and giving the clear aim of the study.
Line 13: replace the word used “ward off” with a scientific term, for example “to prevent…”.
CD14+: you need to delete the space between CD and 14+ and maintaining the same version in whole text, also for the other CDs mentioned.

Lines 17-19: the sentence: “Further, it was investigated…”, isn’t an objective if you express in this way, careful!

Line 28: the word “cocultured” is better to be written “co-cultured”.

Results: you need to replace the correct form for each interleukin and others in whole text, not only in the abstract.

Conclusions: Line 40: replace the word “milieu” with a scientific one, for example “medium or environment”….

1. Introduction: this section needs a reformulation regarding the terminology of CD and others, maintaining the same expression of them.

Aim: this part is very confusing; you need to be more precise and clearer.

2. Materials and methods: Lines from 110-122 need a subtitle to be clear about the description of the work. It is necessary to put a workflow, because is difficult to understand this point.

Line 130: the number "300" given for centrifugation does not have the correct unit behind "g" ... ???

- You need to insert punctuation in long sentences, insert commas where necessary.

Line 137: separate 20 from the unit; [20 mL].

Line 151: “microbeads” - choose another scientific term.

Lines 155-156: put in brackets the name of the device used.

Reformulate the statistical analysis, put SD in brackets, explain also about the significance of p <0.01 regarding to p <0.05.

3. Results: Lines 288-289: this sentence is better to put in the section of materials, not here.

Point 3.1: where is the table 2? p-values are better to put in figure legends and not in the descriptive text, where you need to put your results. It is unclear as a section!

Point 3.2: where is the table 3? The same problem observed here as well as in section 3.1 and other points, such as 3.3., 3.4 and 3.5.

Point 3.3: where are the tables 4,5,6,7 and 8?

Point 3.4: where is the table 9?

Point 3.5: where are the tables 10 and 11?

4. Discussion: this section is very long and contains a lot of confusion. You must clearly discuss your results and using a simple written English. Enrich this section with other references.

Lines 396-397: not clear the English used.

Line 406: check the sentence, you have to discuss your results and not the objectives.

5. Conclusions: reformulate better this section.

Lines 507-508: isn’t a conclusion; it seems like a final discussion.

 Reference suggested: Interferon Crevicular Fluid Profile and Correlation with Periodontal Disease and Wound Healing: A Systemic Review of Recent Data. Int J Mol Sci. 2018 Jun 29;19(7):1908. doi: 10.3390/ijms19071908. PMID: 29966238

Author Response

Reviewer 2:

General comment:

From the title, you expect a lot from this work, but honestly, I found a lot of confusion! It is an interesting subject to treat and the authors need to correct the manuscript as it should, then to be considered for publication. Please find the critical comments as follows:

Critical comments:

Abstract

  1. Lines 16-22: the objectives are not described properly. It is necessary a new reformulation and giving the clear aim of the study.

Response: Respected reviewer, the objectives of the study have been presented with clarity in the revised manuscript

2.Line 13: replace the word used “ward off” with a scientific term, for example “to prevent…”.

Response: respected reviewer we have replaced the word “ward off” with “prevent”.

3.CD14+: you need to delete the space between CD and 14+ and maintaining the same version in whole text, also for the other CDs mentioned.

Response: respected reviewer we have implemented the changes throughout the manuscript.

4.Lines 17-19: the sentence: “Further, it was investigated…”, isn’t an objective if you express in this way, careful!

Response: Respected editor, the word” further investigated” has been removed and rephrased as above advised in the revised manuscript

5.Line 28: the word “cocultured” is better to be written “co-cultured”.

Response: respected reviewer we have changed the word “cocultured” to “co-cultured” throughout the manuscript.

6.Results: you need to replace the correct form for each interleukin and others in whole text, not only in the abstract.

Response:Respected reviewer, we have implemented the changes throughout the manuscript.

7.Conclusions: Line 40: replace the word “milieu” with a scientific one, for example “medium or environment”….

Response: Respected reviewer, we have replaced the word “milieu” with “environment”.

  1. Introduction: this sectionneeds a reformulation regarding the terminology of CD and others, maintaining the same expression of them.

Response:  Respected reviewer, we have implemented the changes in the introduction.

9.Aim: this part is very confusing; you need to be more precise and clearer.

Response: The aim of the study as a part of the introduction has been modified in the revised manuscript

  1. Materials and methods: Lines from 110-122need a subtitle to be clear about the description of the work. It is necessary to put a workflow, because is difficult to understand this point.

Response: Respected reviewer, the materials and methods section has been partitioned into subheadings to enhance ease of reading. The workflow of the study has been presented as a graphical abstract in the revised manuscript.

  1. Line 130:the number "300" given for centrifugation does not have the correct unit behind "g”

Response:   Respected reviewer, we have corrected the unit and rewritten as “300xg”

  1. You need to insert punctuation in long sentences, insert commas where necessary.

Response: the Grammarly software has been applied and the punctuation has been redone in the revised manuscript.

13.Line 137: separate 20 from the unit; [20 mL].

Response:  Respected reviewer, we have Implemented the change.

14.Line 151: “microbeads” - choose another scientific term.

Response: Respected reviewer, sorry to reiterate that this is an appropriate term. We have rechecked in the glossary of medical terms. Advance thanks for agreeing to help us retain this.

15.Lines 155-156: put in brackets the name of the device used.

Response: Respected editor, we have Implemented the change in the revised manuscript.

16.Reformulate the statistical analysis, put SD in brackets, explain also about the significance of p <0.01 regarding to p <0.05.

Response:Respected editor, we have Implemented the change in the revised manuscript.

  1. Results: Lines 288-289:this sentence is better to put in the section of materials, not here.

Response:Respected reviewer, we have  removed the sentence in the revised manuscript.

18.Point 3.1: where is the table 2? p-values are better to put in figure legends and not in the descriptive text, where you need to put your results. It is unclear as a section!

Response: Respected reviewer, sorry for the error, we have added all the tables in the respected positions in the manuscript. We have also removed the p value details from the results section and updated the p values in the tables and table legends.

19.Point 3.2: where is the table 3? The same problem observed here as well as in section 3.1 and other points, such as 3.3., 3.4 and 3.5.

Response: Respected reviewer, sorry for the error, we have added all the tables in the respected positions in the manuscript. We have also removed the p value details from the results section and updated the p values in the tables and table legends.

20.Point 3.3: where are the tables 4,5,6,7 and 8?

Response: Respected reviewer, sorry for the error, we have added all the tables in the respected positions in the manuscript. We have also removed the p value details from the results section and updated the p values in the tables and table legends.

21.Point 3.4: where is the table 9?

Response: Respected reviewer, sorry for the error, we have added all the tables in the respected positions in the manuscript. We have also removed the p value details from the results section and updated the p values in the tables and table legends.

22.Point 3.5: where are the tables 10 and 11?

Response: Respected reviewer, sorry for the error, we have added all the tables in the respected positions in the manuscript. We have also removed the p value details from the results section and updated the p values in the tables and table legends.

  1. Discussion: this section is very long and containsa lot of confusion. You must clearly discuss your results and using a simple written English. Enrich this section with other references.

Response: Respected reviewer, we have shortened the discussion section and enriched the references as per your recommendation

24.Lines 396-397: not clear the English used.

Response: respected reviewer, we have corrected the English language and improved the grammar throught  the manuscript

25.Line 406: check the sentence, you have to discuss your results and not the objectives.

Response: Respected reviewer, we have rechecked and updated the sentence in the revised manuscript

  1. Conclusions:reformulate better this section.

Response: Respected reviewer, we will reformulate the section as suggested

27.Lines 507-508: isn’t a conclusion; it seems like a final discussion.

Response: Respected reviewer, we will reformulate the conclusion and add the below reference in the discussion section wherever relevant

 Reference suggested: “Interferon Crevicular Fluid Profile and Correlation with Periodontal Disease and Wound Healing: A Systemic Review of Recent Data. Int J Mol Sci. 2018 Jun 29;19(7):1908. doi: 10.3390/ijms19071908. PMID: 29966238”

Reviewer 3 Report

Line 22-24  Should be written as one clear sentence:  GCF samples were obtained from periodontitis subjects and healthy individuals. Gingival tissues and blood samples were also obtained from periodontally and systemically healthy subjects. 

Line 24-26 Should be written as one sentence: Cytokine profiles of the GCF samples were analyzed. CD 14+ monocytes were isolated from whole blood and cultured. The cultured monocytes were treated with the GCF of periodontitis patients to observe if they differentiated into macrophages.

Line 39. The conclusion does not give answers to the questions in Objectives of the study. 

Line 113. In Materials and method, there is no explanation on what grade of periodontitis subjects had by new classification. 

Line 130 Instead of 3000g should be 3000 x g- so would not be confused gravity with grams

Line 285. P values are not for presenting descriptive statistics. please do provide more information on your statistic measurements in this study. 

Line 504. The conclusion should first give answers to questions raised as the aims of the study, and then send a message for clinical implications and further studies. The authors should rewrite this section. 

Author Response

Reviewer 3:

Line 22-24  Should be written as one clear sentence:  GCF samples were obtained from periodontitis subjects and healthy individuals. Gingival tissues and blood samples were also obtained from periodontally and systemically healthy subjects.

Response: Respected reviewer, we have presented the data in one sentence as suggested .  

Line 24-26 Should be written as one sentence: Cytokine profiles of the GCF samples were analyzed. CD 14+ monocytes were isolated from whole blood and cultured. The cultured monocytes were treated with the GCF of periodontitis patients to observe if they differentiated into macrophages.

Response: Respected reviewer, we have presented the data in one sentence as suggested .  

Line 39. The conclusion does not give answers to the questions in Objectives of the study. 

Response: Respected reviewer, The conclusions of the revised manuscript has been drafted to answer all the objected of the study

Line 113. In Materials and method, there is no explanation on what grade of periodontitis subjects had by new classification.

Response: Respected reviewer, we have added the details of periodontitis grading in the revised manuscript 

Line 130 Instead of 3000g should be 3000 x g- so would not be confused gravity with grams

Response: Corrected as 300xg. It was mistakenly written as 3000.

Line 285. P values are not for presenting descriptive statistics. please do provide more information on your statistic measurements in this study. 

Response: Provided statistical test details used for data analysis.

  1. Line 504. The conclusion should first give answers to questions raised as the aims of the study, and then send a message for clinical implications and further studies. The authors should rewrite this section. 

Response: respected reviewer , we have reformulated the conclusion according to your suggestion

Round 2

Round 2

General comment: The authors have ameliorated their paper by changing it respecting the comments and suggestions, but to be perfect and finally caring for publication, the present manuscript requires other changes! Please find the new comments as follows:

New comments: - Check the spaces, commas, dots and express the word in vitro in italics: in vitro, put commas before “which” and “while”…

- Line 18: give the long name of GCF.

- Delete the flowchart from the abstract and insert it after the line 139, where it is mentioned for the first time.

- Lines 86 & 483: put reference 11 and 29 inside the respective brackets.

- Line 148: put a space after the number 300 being followed by the unit and replace degrees centigrade with °C, because you have used it throughout the text as symbol.

- Line 162: start the sentence with an uppercase.

- Line 196: put in lowercase the “X” (5x105 cells…).

- Line 215: put the city as you have done in the other cases.

- Line 243: put “µ” instead of microns.

- Line 266: the same as in Line 196 for the “X”.

- Line 271: put in uppercase the “l” for 100 µL.

- Line 277: the same as in Line 196 & 266 for the “X”.

- Line 315: I also suggest adding "Students" before unpaired and "t" in italics.

- Lines 320 & 325: put the dot at the end of the sentences.

- Line 326: I suggest to delete table 2, because you have demonstrated the values of mean ± SD in the columns of Figure 1; put in the figure’s 1 legend the p-values with their respective asterisks (e.g., A. GM-CSF ** p <0.0001).

- Line 333: put in uppercase “it”.

- Line 339: put in uppercase the “figure”.

- Line 348: it is necessary to add “+” after the CD, since in the X axis you added it ...

- Perhaps the graphs in the figure 3 need to be enlarged, in manner to be improved the quality.

- Line 362: put in uppercase the “protein”.

- Line 344: I also suggest here to delete tables 4 and 5 and inserting the p-values into the figure’s 3 legend with the respective asterisk (see Line 326).

- Suggestion: separate Figure 3 into two figures, where the first one including from A to E and new other figure named Figure 4, from F to I, because it is a confusion on the text. Then change the numbers of the other figures as follows.

- Delete also Tables 6 & 7, because the values are reflected in the respective figure.

- Delete Table 8 and put the long name (median fluorescence intensity) of acronym MFI in the legend of figure 5 or in the respective text.

- Line 418: delete Table 9 and put the respective p-values in the legend of figure 6 (see Line 326); put also as long name the “ns” in the legend of the figure.

- Please, delete Tables 10 and 11 and proceed as I suggested above for the other figure legends.

- Line 468: put the comma before "while" and also improve the sentence, because it is not very clear.

- Line 471: also, here improve the phrase.

- Line 489: correct phenotypic with “phenotype”.

- Discussion: You have to shorten again this section and not repeating uselessly the things; please make it as simple as possible.

- Conclusions: from lines 544 to 551 you have repeated your results, perhaps this part will fit better in the discussion section, decide how you see it best; then lines from 551 to 556 are in a sense your conclusions, but it is necessary to improve the construction of sentences in English.

Author Response

Reviewer 2:

General comment: The authors have ameliorated their paper by changing it respecting the comments and suggestions, but to be perfect and finally caring for publication, the present manuscript requires other changes! Please find the new comments as follows:

New comments: - Check the spaces, commas, dots and express the word in vitro in italics: in vitro, put commas before “which” and “while”…

Response: Implemented the suggested changes.

- Line 18: give the long name of GCF.

Response: Given the long name of GCF.

- Delete the flowchart from the abstract and insert it after the line 139, where it is mentioned for the first time.

Response: the flow chart has been deleted from the abstract section and presented in the relevant area after line 139 as recommended

- Lines 86 & 483: put reference 11 and 29 inside the respective brackets.

Response: Implemented the suggested changes.

- Line 148: put a space after the number 300 being followed by the unit and replace degrees centigrade with °C, because you have used it throughout the text as symbol.

Response: Implemented the suggested changes.

- Line 162: start the sentence with an uppercase.

Response: Implemented the suggested changes.

- Line 196: put in lowercase the “X” (5x105 cells…).

Response: Implemented the suggested changes.

- Line 215: put the city as you have done in the other cases.

Response: Implemented the suggested changes.

- Line 243: put “µ” instead of microns.

Response: Implemented the suggested changes.

- Line 266: the same as in Line 196 for the “X”.

Response: Implemented the suggested changes.

- Line 271: put in uppercase the “l” for 100 µL.

Response: Implemented the suggested changes.

- Line 277: the same as in Line 196 & 266 for the “X”.

Response: Implemented the suggested changes.

- Line 315: I also suggest adding "Students" before unpaired and "t" in italics.

Response: Implemented the suggested changes.

- Lines 320 & 325: put the dot at the end of the sentences.

Response: Implemented the suggested changes.

- Line 326: I suggest to delete table 2, because you have demonstrated the values of mean ± SD in the columns of Figure 1; put in the figure’s 1 legend the p-values with their respective asterisks (e.g., A. GM-CSF ** p <0.0001).

Response: Implemented the suggested changes.

- Line 333: put in uppercase “it”.

Response: Implemented the suggested changes.

- Line 339: put in uppercase the “figure”.

Response: Implemented the suggested changes.

- Line 348: it is necessary to add “+” after the CD, since in the X axis you added it ...

Response: Implemented the suggested changes.

- Perhaps the graphs in the figure 3 need to be enlarged, in manner to be improved the quality.

- Line 362: put in uppercase the “protein”.

Response: Implemented the suggested changes.

- Line 344: I also suggest here to delete tables 4 and 5 and inserting the p-values into the figure’s 3 legend with the respective asterisk (see Line 326).

Response: Implemented the suggested changes.

- Suggestion: separate Figure 3 into two figures, where the first one including from A to E and new other figure named Figure 4, from F to I, because it is a confusion on the text. Then change the numbers of the other figures as follows.

Response: Implemented the suggested changes.

- Delete also Tables 6 & 7, because the values are reflected in the respective figure.

Response: Implemented the suggested changes.

- Delete Table 8 and put the long name (median fluorescence intensity) of acronym MFI in the legend of figure 5 or in the respective text.

Response: Implemented the suggested changes.

- Line 418: delete Table 9 and put the respective p-values in the legend of figure 6 (see Line 326); put also as long name the “ns” in the legend of the figure.

Response: Implemented the suggested changes.

- Please, delete Tables 10 and 11 and proceed as I suggested above for the other figure legends.

Response: Implemented the suggested changes.

- Line 468: put the comma before "while" and also improve the sentence, because it is not very clear.

Response: Implemented the suggested changes.

- Line 489: correct phenotypic with “phenotype”.

Response: Implemented the suggested changes.

- Discussion: You have to shorten again this section and not repeating uselessly the things; please make it as simple as possible.

Response: Respected reviewer, we have tried to shorten and simplify the discussion as much as possible

Conclusions: from lines 544 to 551 you have repeated your results, perhaps this part will fit better in the discussion section, decide how you see it best; then lines from 551 to 556 are in a sense your conclusions, but it is necessary to improve the construction of sentences in English.

Response: respected reviewer,the changes have been made and . The English language has been edited in the latter half of the conclusion as per recommendations
